# Wear Behaviors of TiN/WS$_2$ + hBN/NiCrBSi Self-Lubricating Composite Coatings on TC4 Alloy by Laser Cladding

**Kaiwei Liu** [1,2] **, Hua Yan** [1,2,*]**, Peilei Zhang** [1,2]**, Jian Zhao** [1,2]**, Zhishui Yu** [1,2,*] **and Qinghua Lu** [1,2]

[1] College of Materials Engineering, Shanghai University of Engineering Science, Shanghai 201620, China; 15300631239@163.com (K.L.); peilei@sues.edu.cn (P.Z.); zhaojianhit@163.com (J.Z.); luqh@sues.edu.cn (Q.L.)

[2] Shanghai Collaborate Innovation Center of Laser Advanced Manufacturing Technology, Shanghai 201620, China

\* Correspondence: yanhua@sues.edu.cn (H.Y.); yu_zhishui@163.com (Z.Y.); Tel.: +86-21-67791412 (H.Y.); Fax: +86-21-67791377 (H.Y.)

**Abstract:** TiN and WS$_2$ + hBN reinforced Ni-based alloy self-lubricating composite coatings were fabricated on TC4 alloy by laser cladding using TiN, NiCrBSi, WS$_2$, and hBN powder mixtures. Energy-dispersive spectroscopy (EDS), scanning electron microscopy (SEM), X-ray diffractometry (XRD), and optical microscopy (OM) were adopted to investigate the microstructure. The wear behaviors of the self-lubricating composite coatings were evaluated under large contact load in room temperature, dry-sliding wear-test conditions. Results indicated that the phases of the coatings mainly include γ-Ni, TiN, TiNi, TiW, WS$_2$, and TiS mixtures. The average microhardness of the composite coating is 2.3–2.7 times that of the TC4 matrix. Laser cladding TiN/WS$_2$ + hBN/NiCrBSi self-lubricating composite coatings revealed a higher wear resistance and lower friction coefficient than those of the TC4 alloy substrate. The friction coefficient (COF) of the coatings was oscillating around approximately 0.3458 due to the addition of self-lubricant WS$_2$ + hBN and hard reinforcement TiN. The wear behaviors testing showed that the wear resistance of the as-received TC4 was significantly improved by a laser cladding TiN/WS$_2$ + hBN/NiCrBSi self-lubricating composite coating.

**Keywords:** laser cladding; NiCrBSi coating; self-lubricating; microstructure; wear behaviors

## 1. Introduction

TC4 alloy (Ti–6Al–4V) exhibits low density, large specific strength, high Young's coefficient, good anti-corrosive quality, and prominent biocompatibility, which are considered its advantages. Besides, TC4 alloy is extensively applied in the fields of aeronautics and astronautics, petrochemistry, shipbuilding, biomedical, etc. [1] However, for the poor tribological performance of TC4 alloy, components manufactured by TC4 alloy in high load working environments show limited applications. Applying self-lubricating coatings can significantly enhance the anti-wear characteristics on the surface and extend the workpiece service life [2].

Self-lubricating coatings have been widely used to fabricate components operating under extreme conditions, and solid self-lubricating composites exhibit prominent tribology, stabilizing chemical properties, broad temperature range, and pro-environment properties, which are capable of generating prominent self-lubricating effects in friction. Overall, self-lubricating coatings cover matrix, solid lubricant, and auxiliary parts. To form a coating exhibiting smooth smoothness and an appropriate dilution rate and to remedy the defects (e.g., stomata, slag clamping, cracking, and crack occurrence), the thermal expansion coefficient exhibiting prominent wettability and Matrix TC4 alloy ($1.025 \times 10^{-5}$/K$^{-1}$) should be applied. Besides, the melting point reaches below that of the matrix (1538 °C), and the

materials have a low oxygen content. NiCrBSi powder refers to a self-melting alloy powder with a low melting point (1027 °C); the content of oxygen is zero, and the wettability is prominent. Moreover, its thermal expansion coefficient ($1.14 \times 10^{-5}/K^{-1}$) is consistent with that of TC4 alloy [3]. Bowen et al., based on laser cladding, processed the surface of TC4 alloy to prepare an NiCrBSi wear-resistant composite coating. The wear amount was achieved as only 11.4% of the substrate. The increase in the surface microhardness of these coatings led to the enhancement of the wear resistance [4].

TiN has been shown to be an effective enhancement phase for Ti and titanium alloys [5–12]. Its advantages consist of high hardness, prominent wear resistance, and high friction performance. Lubrication phase $WS_2$ could form a lubrication transfer film on the contact surface in the friction, which is critical to decrease the wear rate and coefficient of friction of the coating. The mentioned result was explained as the $WS_2$ being layered between the layer and the layer under Van der Waals force, exhibiting a small shear strength. $WS_2$, as a solid lubricant, shows broad applications [13,14]. Besides, at high temperatures (1000 °C), hBN exhibits high lubrication, and it is a conventional solid lubricant [15–17]. Weng et al., based on laser cladding, delved into the synthesis of the $Co/TiN/Y_2O_3$ composite coatings on TC4 alloy surface; the average microhardness ($1197.9HV_{0.2}$) reached nearly 3–4 times that of the substrate, and the wear rate ($2.525 \times 10^{-4}$ g/min) reached 1/12th of the substrate ($2.995 \times 10^{-3}$ g/min), and the composite coating exhibits prominent wear resistance [5]. Lu et al. clad an Ni60-hBN composite coating on a TC4 board and under the existence of a TiC and $TiB_2$ reinforced phase, the average microhardness of the coating with 10% hBN granule (nearly $1155.32HV_{0.2}$) was approximately threefold that of the substrate (nearly $370HV_{0.2}$) [16].

Laser cladding exploits laser as the energy source, with the matrix metallurgical bonding, heating, and cooling speed. The selection of powder shows few restrictions; the thermal input and distortion are small, the cladding area can be precisely selected, and the process can efficiently achieve automation, environment-related protection, material saving, etc. However, the decomposition and evaporation of sulfide as a lubrication phase occurs, limiting it to be industrially applied in laser clad self-lubricating composite coatings. An in-situ method to synthesize composite coatings refers to a good solution to this problem.

In this study, based on laser cladding technology, a self-lubricating composite coating of TiN, TiNi, TiW, TiS, and an hBN lubrication phase was precast on TC4 alloy substrate. $WS_2$ can easily decompose at high temperatures, and the Ti element exhibits high chemical affinity, which helps synthesize the TiS lubrication phase. A prominent composite self-lubricating coating was achieved with TiS and hBN lubrication, and the synergy was achieved between rigid titanium intermetallic compounds.

## 2. Experimental Procedure

### 2.1. Preplaced Layer Preparation

First, this study prepared TC4 alloy substrate in a 100 mm × 50 mm × 10 mm rectangular parallelepiped sample. Subsequently, the surface was removed by sanding for the removal of the oxide layer. The nickel-base alloy coating powder (Ni60, powder size: 45–105 μm), TiN (particle diameter: 1–3 μm), hBN (5–7 μm), and tungsten disulfide (particle size: 0.5–1 μm) were selected. Table 1 lists the chemical compositions of the Ni60 and TC4 substrate. For determining the coating microstructure variations, the coating material refers to a mixture of a variety of hBN and $WS_2$ content and quantitative TiN and Ni60. The powder for the mixed coating is presented in Table 2. To ensure homogeneous mixing, the mixture was milled in a planetary ball mill for 4 h. A mixed powder for laser cladding was preset on the TC4 substrate. Overall, the method to bond applies to the preparation of preset layers. The pre-lay layer thickness was regulated at 1.5 mm. Finally, the prefabricated coating was placed in a drying oven at 100 °C for 2 h.



**Table 1.** Chemical composition of TC4 substrate and NiCrBSi powder.

| Material | Element Composition (wt.%) | | | | | | | | | | | | | |
|---|---|---|---|---|---|---|---|---|---|---|---|---|---|---|
| | **C** | **Cr** | **Si** | **Al** | **Mo** | **V** | **Mn** | **N** | **O** | **Fe** | **H** | **B** | **Ti** | **Ni** |
| Substrate | 0.30 | – | – | 6.30 | – | 4.20 | – | 0.05 | 0.20 | – | 0.01 | – | Bal. | – |
| NiCrBSi | 0.72 | 15.1 | 4.10 | – | 0.02 | – | 0.01 | – | – | 3.77 | – | 3.30 | – | Bal. |

**Table 2.** The mixtures composition for coatings.

| Number | Mixtures Composition (wt.%) | Mixtures Composition (v.%) | Mass/g |
|---|---|---|---|
| Coating 1 | NiCrBSi + 30% TiN + 15% $WS_2$ | 62.8% NiCrBSi + 27.3% TiN + 9.9% $WS_2$ | 50 |
| Coating 2 | NiCrBSi + 30% TiN + 15% hBN | 50.1% NiCrBSi + 22.1% TiN + 26.8% hBN | 50 |
| Coating 3 | NiCrBSi + 30% TiN + 5% $WS_2$ + 10% hBN | 54.5% NiCrBSi + 23.8% TiN + 2.8% $WS_2$ + 18.9% hBN | 50 |
| Coating 4 | NiCrBSi + 30% TiN + 7.5% $WS_2$ + 7.5% hBN | 56.3% NiCrBSi + 24.5% TiN + 4.4% $WS_2$ +14.8% hBN | 50 |
| Coating 5 | NiCrBSi + 30% TiN + 10% $WS_2$ + 5% hBN | 58.3% NiCrBSi + 25.3% TiN + 6.1% $WS_2$ + 10.2% hBN | 50 |

### 2.2. Laser Cladding

For the IPGYLS-5000 fiber laser system (IPG, Parma, Italy), laser cladding based on an innovative gas protection device was conducted. In the preliminary studies, the process parameters to synthesize the optimal laser-clad NiCrBSi powder on the TC4 substrate consisted of a laser output power of 3.0 kW, a scanning speed of 15 mm·s$^{-1}$, as well as a spot diameter of 2 mm. This study selected argon gas at a flow rate of 10 L·min$^{-1}$ as the shielding gas. Under the eliminated impact of the laser cladding parameters, the microstructure variations of the coating can be more effectively clarified. Thus, all the sample parameters taken in this study were identical. The shielding gas was turned on 3 s before the laser transmitter operated; then, it was turned off 3 s after the end of the program. The single pass and overlap coatings were air-cooled under an overlap ratio of 40%. With a single coat, X-ray diffraction and SEM analysis were conducted. An overlapping coating was used for wear testing.

### 2.3. Microstructural Features

With a wire cutter, the single-coated sample was cut in a vertical section, and the sample underwent the treatment with a standard metallographic method. Lastly, the sample was etched with Keller reagent (volume ratio HF:HCl:$H_2O$ = 1:3:46) for 20 s. By X-ray diffraction (XRD, X'Pert Pro, PANalytical, EA Almelo, The Netherlands), the crystalline phase of the coating was determined. Under a scanning electron microscopy (SEM, S-3400, Hitachi, Tokyo, Japan), this study characterized the wearing morphology of the coating. With an HXD-1000TMSC/LCD Vickers hardness tester (Taiming, Shanghai, China), the microhardness was repeatedly determined in the depth direction of the coating and then averaged.

### 2.4. Tribological Studies

Tribological studies were conducted with a pin-and-disc friction and wear tester (UMT-3, Bruker, Baden-Württemberg, Germany) as per ASTM G99 standard at room temperature (24 °C) in the ambient atmosphere. Figure 1 illustrates its schematic diagram. All the wear test specimens had a diameter of 15 mm and thickness of 5 mm during a 30-min dry slip friction test in which the parameter was 5.0 kg normal load, the speed was 100 rpm, and the grinding radius was 3 mm. The rubbing pair was a hard WC ball with the diameter of 9.5 mm, and a ball was replaced when a respective sliding friction test was completed. Then, the friction coefficient (COF) was logged during the tests. Under a three-dimensional microscopic system of super depth of field (VHX-5000, KEYENCE, Osaka, Japan), the wear morphology of coatings and substrate was performed. The wear rate is expressed by the wear expressions below:

$$W = \frac{V}{LS} \tag{1}$$

where *W*, *V*, *L*, and *S* respectively denote the specific wear rate (mm$^3$·N$^{-1}$·m$^{-1}$), wear volume (mm$^3$), normal load (N), and sliding distance (m) [18].

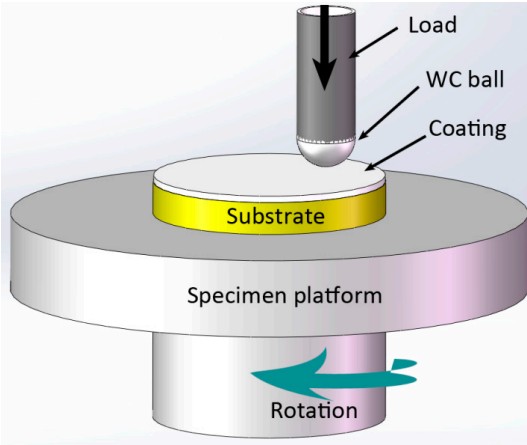

**Figure 1.** Pin-and-disc friction and wear tester schematic diagram.

## 3. Results and Discussion

### 3.1. Microstructural Characterization

X-ray diffraction (XRD) analysis regions of the five samples are presented in Figure 2. The laser cladding process exhibits a non-equilibrium state of rapid solidification, thereby resulting in multiphase coexistence, and the diffraction peaks of the phase overlap the equilibrium state, so the laser cladding coating is difficult to recognize at all phases. All diffraction peaks are referenced based on the Joint Committee on Powder Diffraction Standards (JCPDS). As revealed from the index results, the main diffraction peak of the coating complies with the following JCPDS cards: TiN was 01-087-0632, TiW was 03-065-6898, TiS was 00-051-1329, and TiB was 00-006-0641. As revealed from the analysis of the XRD pattern, the coating largely consisted of a mixture of $\gamma$-Ni, TiN, TiNi, TiW, $WS_2$, and TiS. The mixture (e.g., TiNi, TiW, and TiS) was formed by the vast Ti element generated by a partially melted matrix, and the Ni, S, and W elements were generated by the additive deposited metals that reacted with each other in the molten pool. The diffraction peaks were clearly observed in the coating with TiN added, suggesting that the high chemical durability of TiN [19]. TiN had a diffraction peak at 36.7° (111), 42.63° (200), and 61.9° (220), demonstrating that the composite coating consisted of mixed crystal grains with orientation components. Since TiN is a face-centered cubic lattice structure with a closely packed (111) direction, the energy required to grow TiN in this direction was low [20]. Accordingly, the preferred orientation of TiN is (111) [21]. As suggested from the XRD analysis, the diffraction peak intensity of coating 2 at 39.554° was the strongest, and the diffraction peak intensity of coating 3, coating 4, and coating 5 increased with the increase in $WS_2$ phase. It was because the intermetallic compound TiW was formed, and the diffraction peak was present at 39.554°. In the course of laser cladding, although $WS_2$ was easily decomposed (decomposition temperature at 510 °C), the experiment used a certain method to save or regenerate into $WS_2$. The diffraction peak of the lubricating phase hBN was clearly identified in coating 1.

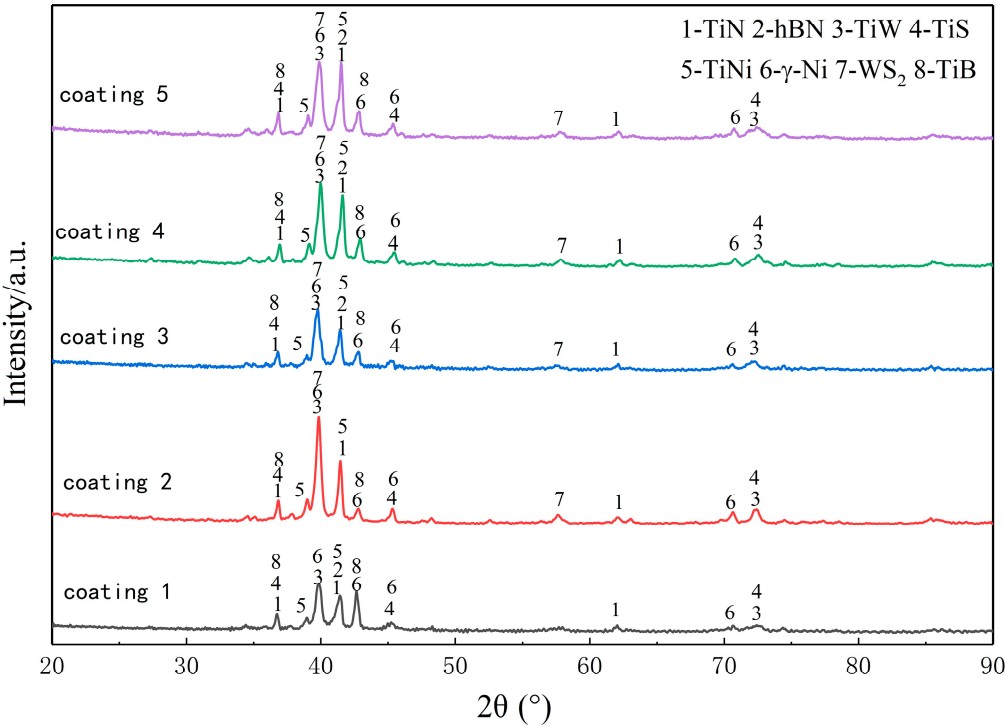

**Figure 2.** XRD pattern of composite coating.

The cross-section microscopic morphology of composite coatings was characterized in Figure 3, containing a continuous matrix, dendrites phase, bar phase, and branches crystals, and the phase structure was more refined (the grain size was smaller). Figure 3a shows an overview cross-sectional feature of a composite coating. This figure suggests that the coating and substrate exhibited a metallurgical combination. Figure 3b shows the upper region of the composite coating, and the middle part of the composite coating is depicted in Figure 3c. Contrasting the upper, middle, and bottom parts of the coating, the upper region had a larger microstructure than the intermediate region, since the degree of supercooling and the cooling rate of the upper part of the composite coating surpassed the middle region of the coating, and the critical nucleation radius of the upper part of the coating was larger, thereby achieving a larger microstructure of the upper part of the coating. The composite coating had fine dendrites distributed in the region close to the bonding zone, as indicated in Figure 3d, and the bonding region showed planar crystal and columnar crystal growth, which was significantly different from the microstructure of the upper and middle regions of the coating. Since the bonding zone was close to the substrate, the temperature gradient $G$ ($G$ as the liquid phase temperature gradient) close to the substrate region in the initial molten pool solidification phase was significantly larger than zero, the crystallization velocity $R$ was close to zero, and the value of $G/R$ was close to infinity. The planar crystal preferentially grew outward from the substrate. The subsequent temperature rise of the substrate caused $G$ to decrease and $R$ to increase, and then the value of $G/R$ decreased, facilitating the growth of dendrites [21]. Accordingly, fine dendrites were distributed near the bonding zone.

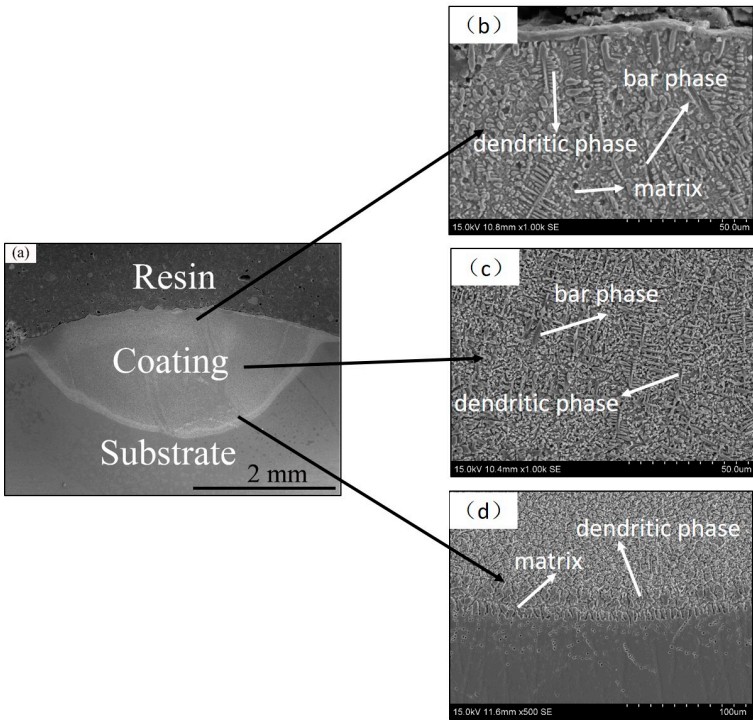

**Figure 3.** SEM topography of coatings: (**a**) macroscopic morphology, (**b**) upper part, (**c**) middle part, (**d**) bottom part.

The chemical composition of the different phases in the coating (Figure 4) was determined with an energy-dispersive spectroscopy (EDS) spectrometer. The distribution of the elements is listed in Table 3. The EDS study outcomes and XRD analysis suggested that the dendritic phase A in Figure 4 primarily contained Ti (49.50%) and N (49.19%) elements, which were presumed as TiN; the gray phase B primarily covered Ti and Ni elements. It is speculated that the two phases are TiNi; the acicular phase C is mainly composed of Ti (72.09%) and S (9.91%) elements, which is supposed to be TiS; the layered substance D in the figure is primarily composed of W (13.72%); the composition of S (15.62%) is presumed to be WS$_2$; the phase F primarily convers Ti (46.35%) and W (42.28%) elements, as presumed to be TiW. Therefore, it is revealed that the Ni-based self-lubricating composite coating covers both a hard phase (e.g., TiN, TiNi, and TiW) and a TiS lubricating phase formed in situ. The hBN content is small and cannot be identified in EDS.

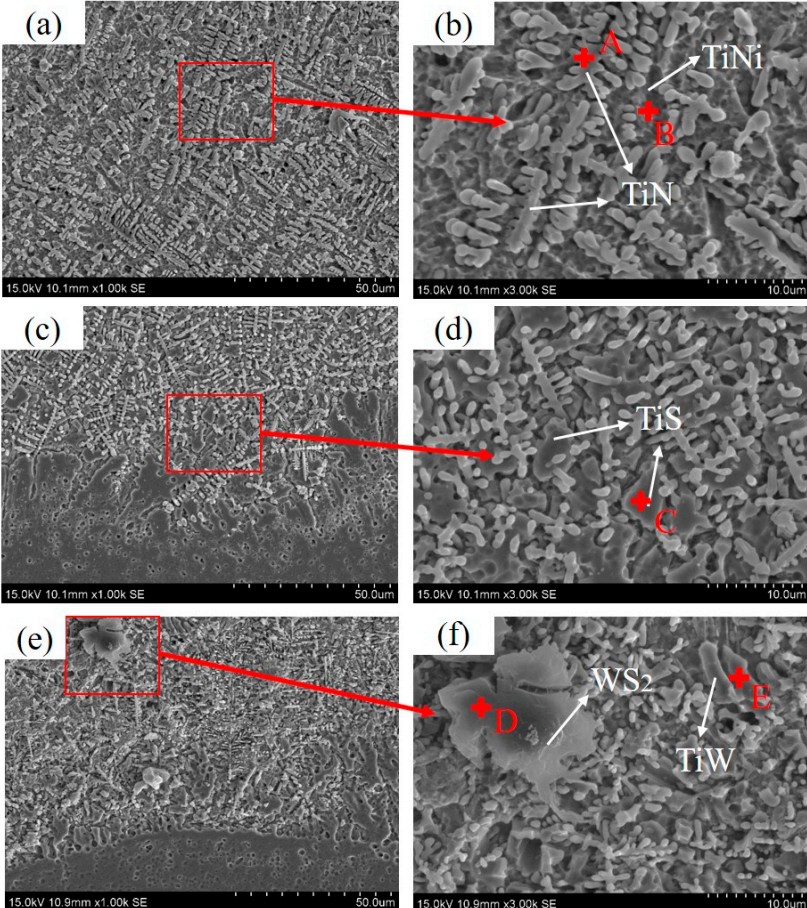

**Figure 4.** SEM topography of the chemical composition of the different phases in the coatings: (**a**) the middle area of coating 2; (**b**) a enlarge picture of Figure 4a; (**c**) the Bottom area of coating 3; (**d**) a enlarge picture of Figure 4d; (**e**) the upper area of coating 5; (**f**) a enlarge picture of Figure 4e.

**Table 3.** Chemical compositions of the phases exhibiting a range of morphological characteristics in the coatings.

| Zones | Elements (at.%) | | | | | | | | | Phase |
|---|---|---|---|---|---|---|---|---|---|---|
| | Ti | Al | V | Ni | Cr | Si | N | S | W | |
| A | 49.50 | 0.69 | 0.62 | – | – | – | 49.19 | – | – | TiN |
| B | 32.13 | 3.61 | 1.61 | 34.82 | 15.41 | 12.42 | – | – | – | TiNi |
| C | 72.09 | – | 1.54 | 10.61 | 2.78 | 3.07 | – | 9.91 | – | TiS |
| D | 55.07 | 5.14 | 1.42 | 7.47 | 1.56 | – | – | 15.62 | 13.72 | WS$_2$ |
| E | 46.35 | 4.26 | 7.11 | – | – | – | – | – | 42.28 | TiW |

*3.2. Microhardness Test*

Figure 5 presents the microhardness of five coatings (i.e., coatings 1, 2, 3, 4, and 5) along the depth direction of the coating cross-section, demonstrating that the microhardness of the five coatings was basically consistent along the depth of the layer and primarily covered the matrix, the heat-affected area, and the cladding layer. Furthermore, the microhardness of the coating displayed a gradual increase along the mentioned direction. Moreover, the microhardness of the cladding layer remained almost unchanged and was relatively stable, demonstrating that the cladding layer microstructure exhibited overall homogeneity. The average microhardness of the five coatings respectively reached 860 ± 12.8HV$_{0.2}$ (coating 1), 842 ± 16.4HV$_{0.2}$ (coating 2), 870 ± 14.5HV$_{0.2}$ (coating 3), 1006 ± 6.6HV$_{0.2}$ (coating 4), and 926.5 ± 19.8HV$_{0.2}$ (coating 5), respectively. It was about 2.3–2.7 multiple that of

the base TC4 alloy (about $370HV_{0.2}$). Since the laser cladding could strengthen the fine grain, the microhardness of the cladding layer was enhanced. The hard/soft phase composite structure exhibited high hardenability and plastic reserve capacity, which led to the enhancement of the hardness of the cladding layer. A hard-reinforcing phase (e.g., TiNi and TiB) was dispersed in the coating to produce dispersion strengthening, which significantly enhanced the hardness of the coating. As impacted by the existence of the vast lubricating phase TiS in the Ni-based self-lubricating composite matrix, the hardness of the coating was reduced. Coating 2 had a soft texture and low hardness for the considerable lubricating phases $WS_2$ and TiS.

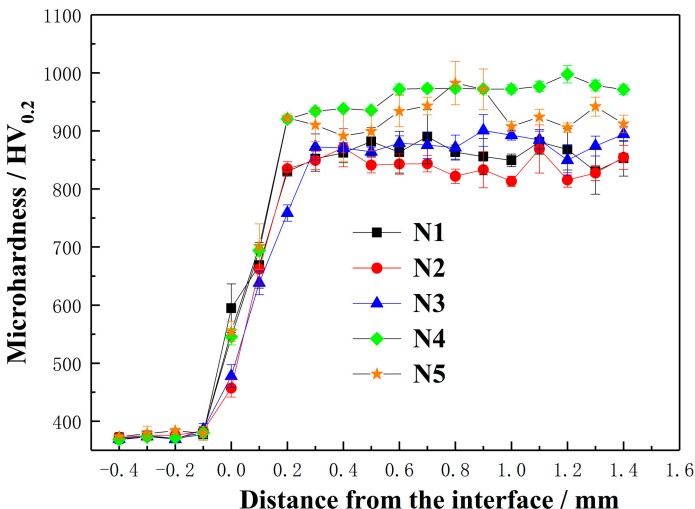

**Figure 5.** The microhardness of coatings along the depth direction of the cross-section.

### 3.3. Tribological Behavior

### 3.3.1. Analysis Friction Coefficient

Figure 6 shows the wear of substrate and coating after a wear behavior test was performed for 30 min at ambient temperature. All the experimental parameters are marked in the friction and wear experiment (Section 2.4 presents the specific test parameters). Under the introduction of NiCrBSi, TiN, hBN, and $WS_2$, the abrasion resistance of the coating with the microstructure synergistically and the addition of the reinforcing phase TiN significantly enhanced the hardness of the coating. Moreover, different levels of lubricating phase led to different degrees of lubrication. The coefficient of friction (COF) curves of five coatings are presented in Figure 6. The average friction coefficient and wear rate are presented in Figure 7. Figure 7 suggests that the coefficient of friction of the coating can be classified into two moments, i.e., the initial (3 min after the start) and stable wear phase. The coefficient of friction of the coating slightly fluctuated after reaching a stabilization wear stage. Compared with the average coefficient (i.e., coatings 1 and 2), it was verified that hBN and $WS_2$ can enhance the coating lubricity at ambient temperature. Coating 2 (0.4291) contained a plentiful texture of TiS and $WS_2$, which was soft, and the hardness was low. Although the friction could be reduced in the early stage of wear, the friction coefficient rose faster, as the lower hardness lubrication phase is difficult to preserve. Thus, it had significantly less internal stress and better toughness, more effectively preventing the ball from being immersed in the coating. Besides, the self-lubricating phase TiS generated in situ in the self-lubricating composite coating could facilitate the abrasion resistance and friction reduction of the composite coating. hBN significantly enhanced the hardness and wear resistance of the coating at the beginning of the experiment, so a relatively stable coefficient of friction was achieved before coating 1 for 200 s. Besides, the composite self-lubricating composite coatings 3 (0.3458), 4 (0.3948), and 5 (0.4072) suggested a further decrease in the average coefficient of friction. This result was achieved because the self-lubricating composite coating contained hard phases TiN, TiW, TiNi and lubricating phases

TiS, WS$_2$, and hBN. Though the self-lubricating composite, coatings 3, 4, and 5 had a stable friction coefficient after reaching a stable wear stage, while the friction coefficients of coatings 1 and 2 increased with the prolonging of the wear time. The stable growth period could effectively reflect their wear resistance well.

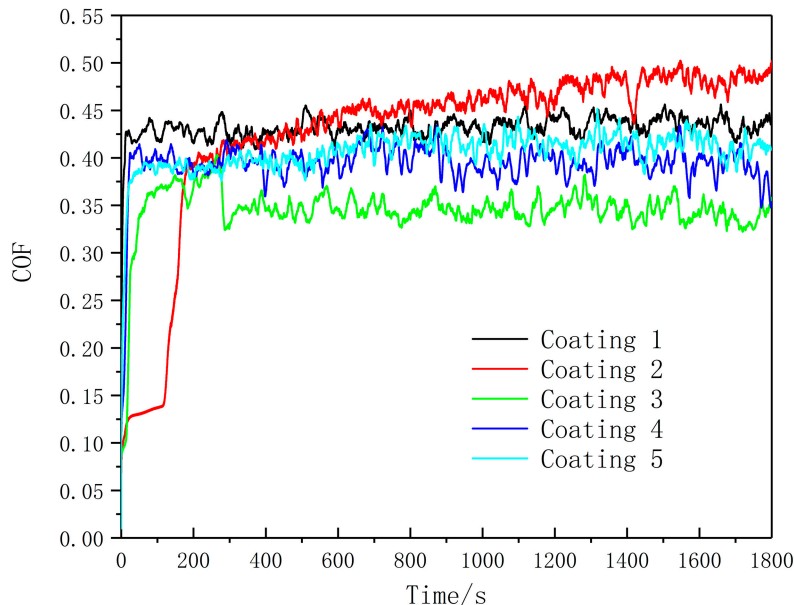

**Figure 6.** The variations in friction coefficient with the sliding time of coatings.

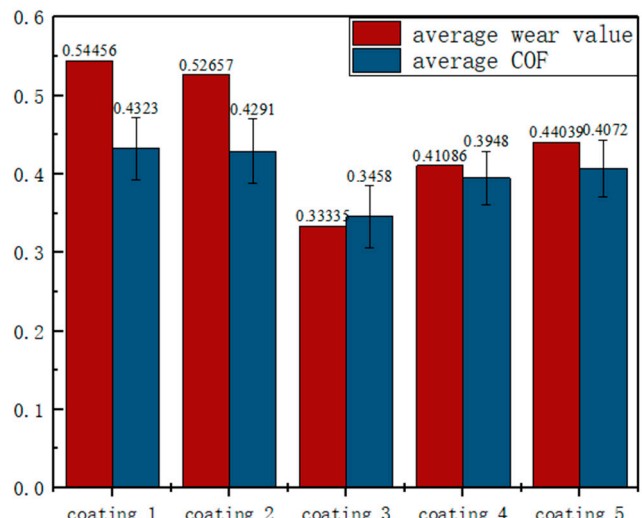

**Figure 7.** Average friction coefficient and wear rates of coatings.

It is noteworthy that for coating 3, the coefficient of friction of the coating was nearly identical to that of the other coatings at the beginning, whereas the friction coefficient of the coating decreases and remains stable for nearly 3 min. This result was achieved as the surface wear behavior; the lubricating phase in the coating can form a lubricating film that protects the coating from wear. Moreover, with the increase in WS$_2$ content, the wear resistance decreased in coatings 3, 4, and 5. This was because the TiS lubricating phase in the self-lubricating composite coating 3 formed a lubricant transfer film on the worn surface in the sliding direction during friction, thereby effectively enhancing the wear resistance of the coating. The self-lubricating composite coating 4 might be due to the lack of sufficient lubrication source, so the wearing resistance of the coating could not be effectively enhanced; as a result, a wider

and deeper wear scar was formed. The self-lubricating composite coating 5 was due to the existence of defects in the tissue, which resulted in a wide and deep wear scar.

According to all tribology test results, the mentioned coatings had significantly lower wear rates than the substrate. Under the addition of NiCrBSi, TiN, hBN, and $WS_2$, the variation in the wear resistance of the coating complied with the evolution of the microstructure. According to Section 3.1, there was a hard phase between coating 1 (Ti–Ni intermetallic, TiB, TiN, and hBN) and coating 2 (Ti–Ni metal, TiW, TiS, TiN, and $WS_2$). In addition, there was a hard phase between coating 3 (Ti–Ni metal, TiN, TiB, TiW, TiS, and $WS_2$), coating 4 (Ti–Ni metal, TiN, TiW, and TiS), and coating 5 (Ti–Ni metal, TiN, TiW, TiS, and $WS_2$). TiN and in situ Ti–Ni intermetallic compounds promoted resistance to the wear of coatings. Lubricating TiS endowed coatings with better wear resistance and the down-regulation of friction [21].

### 3.3.2. Analysis of Wear Principle

For the research of the friction principle and the additive operation of the coating in the friction course, the wear scar morphology and wear volume are shown by the 3D microscope method of extreme field depth in Figure 8, and Figure 9 presents SEM photos of the wearing morphological characteristic of the samples of the coatings. It was suggested that coating 3 exhibited the optimal tribology behavior (i.e., the lowest average friction coefficient and wear rate). The scar morphology of coating 3 was the shallowest and the most flat, which only had a small of spalling crater and shallow plough, so the major wearing principle of coating 3 was fatigue wear. Figure 8a reveals a mass of wear debris and the deepest plough on the coating surface, since hBN hard phase particles were exposed on the surface of coating, and the coarse particles were pushed and peeled off by the friction pair in the sliding. Thus, they plastically flowed and plowed a groove in coating 1, which was considered to be typical abrasive wear. In the event of coating 2, the additive phase of $WS_2$ reduces the hardness of the coating and makes the coating fragile. In the course of friction, the self-lubricating phase on the soft surface was easily peeled off, generating spalling craters, as the picture presents in coating 2 (Figure 9b), and causing adhesive wear. Followed by coating 4, it achieved both deep plough and spalling crater, and the wear profile was irregular. The wear process was closely related to the molecular force and frictional heat between the surfaces, which was the result of the synergy between adhesive and abrasive wearing. Coating 5 had considerable wear debris pressed into the friction surface under load to cause an indentation, and the surface of the plastic material was extruded into layered or scale-like exfoliated debris. As revealed from the mentioned conclusions, this friction and wear test of the surface of the composite coating exhibited scratches, peeling, and plastic deformation. The wearing principle of the coating refers to adhesive and abrasive wearing. Coating 3 has the optimal resistance to wear.

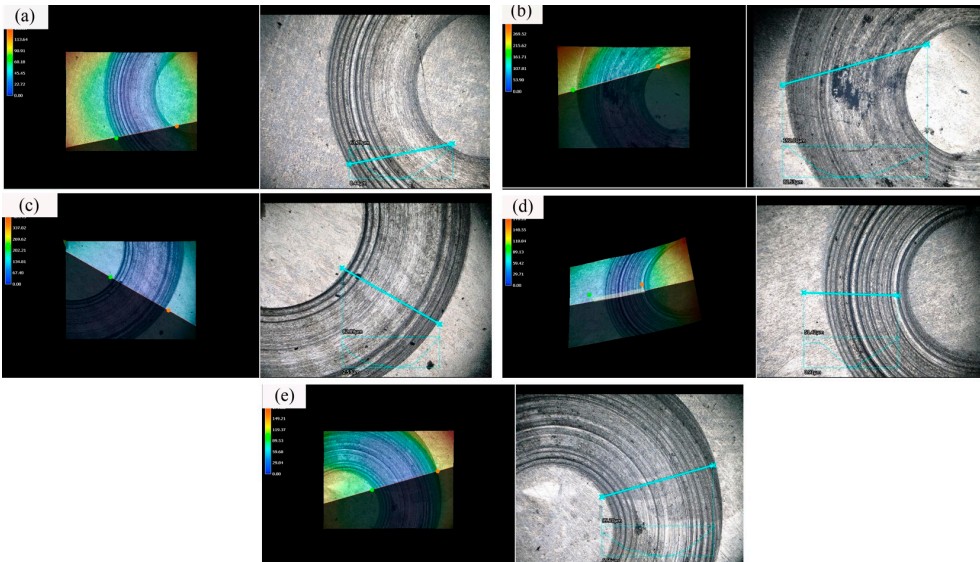

**Figure 8.** Wear scar morphology and wear volume by a 3D microscopic system of extreme field depth. (**a**) Coating 1, (**b**) coating 2, (**c**) coating 3, (**d**) coating 4, (**e**) coating 5.

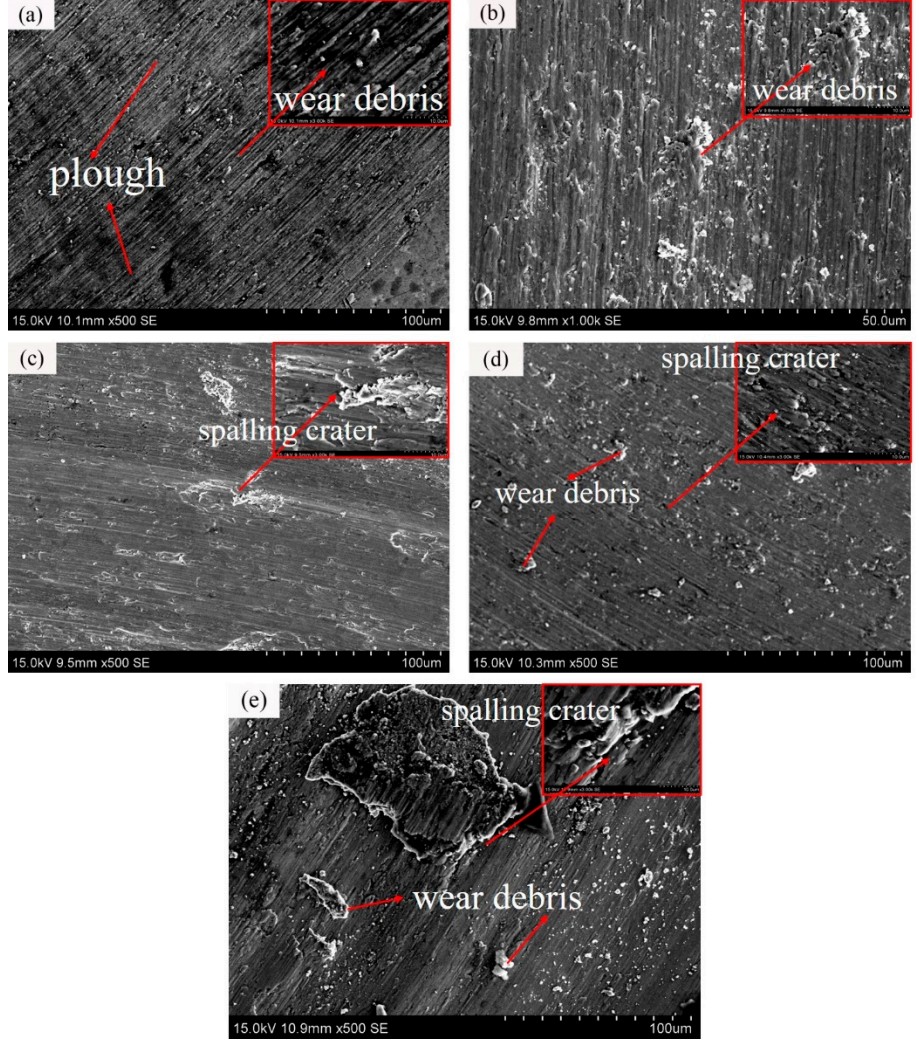

**Figure 9.** SEM images of the surface wear morphology of the different coated samples: (**a**) coating 1, (**b**) coating 2, (**c**) coating 3, (**d**) coating 4, (**e**) coating 5.

## 4. Conclusions

- This study, based on laser cladding, well synthesized Ni-involved metal matrix self-lubrication coatings (e.g., reinforced phase TiN, hBN, and solid-lubricant phase $WS_2$) on a TC4 alloy surface. $WS_2$, hBN, TiS, TiW, TiN, TiNi, and TiB were synthesized in the coating.

- In the solidifying course, the lubricating phases $WS_2$ and hBN react with the matrix and substrate to form the lubricating phases TiS and TiW, the reinforcing phases TiNi and TiN, and retain a certain amount of $WS_2$ and hBN.

- As lubricating and reinforcing phases are added, the composite coating showed enhanced resistance to wear and microhardness. Coating 4 (NiCrBSi-30% TiN-5% $WS_2$-10% hBN) achieved a hardness over $1005.98HV_{0.2}$, which was almost 2.68 times the hardness of the substrate. The minimal coating friction was 0.3458 in pin-and-disc friction and wearing testing elements under ambient temperature; therefore, the lowest wear rate is 0.3335, and the major wearing principle was abrasive and adhesive wearing.

**Author Contributions:** Conceptualization, J.Z. and P.Z.; methodology, K.L. and H.Y.; date analysis, Z.Y. and Q.L.; data curation, K.L.; writing—original draft preparation, H.Y.; writing—review and editing, K.L. and Z.Y. All authors have read and agreed to the published version of the manuscript.

**Funding:** The present study was financed by the Aid for Xinjiang Science and Technology Project (2019E0235), Karamay Science and Technology Major Project (2018ZD002B), Local College Capacity Building of Shanghai Science and Technology Committee Innovation Programs (19030501300), Shanghai Science and Technology Committee Innovation Grant (17JC1400600, 17JC1400601), National Natural Science Foundation of China (51405288, 51605276).

**Conflicts of Interest:** The authors declare no conflict of interest.

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
