# Peer review of "Wear Behaviors of TiN/WS2 + hBN/NiCrBSi Self-Lubricating Composite Coatings on TC4 Alloy by Laser Cladding"

_coatings, doi:10.3390/coatings10080747_

Round 1
Reviewer 1 Report
1. This manuscript is not well organized and full of writing errors. The title should be changed. What is “wear resistant” of laser cladding?
2. What are the assumptions in wear testing?
3. Please check this sentence: Section 3.3.1 analysis Friction coefficient. It is difficult for me to read. Please rewrite the entire so that I can understand the language. I will be happy to give my comments further.
Author Response
Dear Editors and Reviewers,
Thanks very much for your supervision of the reviewing process of our manuscript ID (No.: coatings-862496). We have revised the manuscript “Wear resistant of laser cladding Ti-based and WS2+hBN self-lubricating composite coating” based on comments and suggestions from the Reviews and Editors. Those comments are all valuable and very helpful for revising and improving our paper, as well as the important guiding significance to our researches. We have studied comments carefully and have made correction which we hope meet with approval. To make the reviewer comments available to all of the authors, we have repeated the comments from each of the reviewers below (in a different font). We have responded to the comments by each reviewer where appropriate and have indicated the changes made in the manuscript. We also highly appreciate the reviewers’ carefulness, conscientiousness, and the rich knowledge on the relevant research fields, since he/she has given me a number of beneficial suggestions. Below are detailed responses to comments from the editors and reviewers. Revised portion are marked in red in the paper. The page and line numbers refer to our revised manuscript submitted. The main corrections in the paper and the responds to the reviewer’s comments are as flowing.
Kind regards,
Kaiwei Liu, Hua Yan, Peilei Zhang, Jian Zhao, Qinghua Lu, Zhishui Yu
Reviewer Comments:
Reviewer #1:
- This manuscript is not well organized and full of writing errors. The title should be changed. What is “wear resistant” of laser cladding?
Thanks for the reviewer’s suggestion for revising the section of title in our manuscript. The suggestion put forward by the reviewer are of great significance. This is an error of grammer, we could modify to “Wear behaviors of TiN/WS2+hBN/NiCrBSi self-lubricating composite coatings on TC4 alloy by laser cladding”
- What are the assumptions in wear testing?
It is really a great suggestion as reviewer pointed out. Thus, we have added the factor into the experimental procedure about the assumptions in tribological behavior. The modifications are as follows:
(L. 112, page 3) Tribological studies were conducted with a pin-and-disc friction and wear tester (Bruker/UMT-3, Baden-Württemberg, Germany) as per ASTM G99 standard at room temperature(24°C) in the ambient atmosphere. Fig.1 illustrates its schematic diagram. All the wear test specimens were with diameter 15 mm and thickness 5 mm during 30min Dry slip friction test in which the parameter was 5.0 kg normal load, the speed was 100rpm and the grinding radius was 3mm. The rubbing pair was a hard WC ball with the diameter of 9.5mm, and a ball was replaced when respective sliding friction test was completed. Then, the COF was logged during the tests.
- Please check this sentence: Section 3.3.1 analysis Friction coefficient. It is difficult for me to read. Please rewrite the entire so that I can understand the language. I will be happy to give my comments further.
We are very sorry for the miss. We aimed to describe a way to fabricate the coating. We revise the whole section as follows.
(L. 213, page 7) Fig. 5 shows the wear of substrate and coating after wear behavior test was performed for 30 min at ambient temperature. All experimental parameters are marked in the friction and wear experiment (section 2.4 presents specific test parameters).Under the introduction of NiCrBSi, TiN, hBN and WS2, the abrasion resistance of the coating with the microstructure synergistically, and the addition of the reinforcing phase TiN significantly enhanced the hardness of the coating. Moreover, different levels of lubricating phase led to different degrees of lubrication. The Coefficients of Friction (COF) curves of five coatings are presented in Fig. 5. The average friction coefficient and wear rate are presented in Fig. 6. Fig. 5 suggests that the coefficient of friction of the coating can be classified into two moment, i.e., the initial (3 min after the start) and stable wear phase. The coefficient of friction of the coating slightly fluctuated after reaching a stabilization wear stage. Compared with the average coefficient (i.e., coating 1 and 2), it was verified that h-BN and WS2 can enhance coating lubricity at ambient temperature. The coating 2(0.4291) contained plentiful texture TiS and WS2, which was soft, and the hardness was low. Though the friction could be reduced in the early stage of wear, the friction coefficient rose faster as the lower hardness lubrication phase is difficult to preserve. Thus, it had significantly less internal stress and better toughness, more effectively preventing the ball from being immersed in the coating. Besides, the self-lubricating phase TiS generated in-situ in the self-lubricating composite coating could facilitate the abrasion resistance and friction reduction of the composite coating. hBN significantly enhanced the hardness and wear resistance of the coating at the beginning of the experiment, so a relatively stable coefficient of friction was achieved before the coating 1 for 200s. Besides, the average coefficient of friction of the composite self-lubricating composite coatings 3 (0.3458), coating 4 (0.3948) and coating 5 (0.4072) decreased further. This result was achieved because the self-lubricating composite coating contained hard phases TiN, TiW, TiNi and lubricating phases TiS, WS2 and hBN. Though the self-lubricating composite coatings 3, 4 and 5 had a stable friction coefficient after reaching a stable wear stage, the friction coefficients of the coatings 1 and 2 increased with the prolonging of the wear time. The stable growth period could effectively reflect their well wear resistance.
It is noteworthy that for the coating 3, the coefficient of friction of the coating was nearly identical to that of the other coatings at the beginning, whereas the friction coefficient of the coating decreases and remains stable for nearly 3 min. This result was achieved as the surface wear behavior, the lubricating phase in the coating can form a lubricating film that protects the coating from wear. Moreover, with the increase in WS2 content, the wear resistance decreased in Coatings 3, 4 and 5. This was because the TiS lubricating phase in the self-lubricating composite coating 3 formed a lubricant transfer film on the worn surface in the sliding direction during friction; thus, the wear resistance of the coating was effectively improved. The self-lubricating composite coating 4 might be due to the lack of sufficient lubrication source, so the wearing resistance of the coating could not be effectively enhanced, as a result, a wider and deeper wear scar was formed. The self-lubricating composite coating 5 was due to its existence defects in the tissue, which resulted in a wide and deep wear scar.
According to all tribology test results, the mentioned coatings had significantly lower wear rates than the substrate. Under the addition of NiCrBSi, TiN, hBN and WS2, the variation in the wear resistance of the coating complied with the evolution of the microstructure. According to Section 3.1, there was a hard phase between coating 1 (Ti-Ni intermetallic, TiB, TiN, hBN) and coating 2 (Ti-Ni metal, TiW, TiS, TiN, WS2). In addition, the hard phase, the coating 3 (Ti-Ni metal, TiN, TiB, TiW, TiS, WS2), the coating 4 (Ti-Ni metal, TiN, TiW, TiS) and the coating 5 (Ti-Ni metal, TiN, TiW, TiS, WS2). TiN and in-situ Ti-Ni intermetallic compounds promoted resistance to wear of coatings. Lubricating TiS endowed coatings with better wear resistance and the down-regulation of friction [21].

Reviewer 2 Report
This work deals with the use laser cladding technology for elaborating composite coatings, exhibiting self-lubricating properties, onto a two-phase Ti alloy. In principle, this issue is of significant engineering importance and suitable for Coatings. However, I cannot propose the publication of the article in its present form, since it suffers at several points:
- What was the volume of the pre-deposited powder mixture ?
- Please revise the content in parenthesis (L. 100, page 3) and the rotational speed (L. 110, page 3): volume ratio HF:HCl:H2O = 1:3:46 and 100 rpm.
- Equation (1), L. 115-116, page 3, does not appear correctly.
- Fig, 2 should be moved earlier and closer to the relevant text. Moreover, the red-coloured marks in (b)), (c) and (d) do not help the reader to follow the discussion. Especially, the marks on (c) that are combined with the data given in Table 2, are not at all clear. Moreover, the low magnification of the image does not allow the reader to understand differences of the microstructure features that could be related to differences of the elemental composition.
- The values of microhardness that are given with a precision of two decimals are rather confusing. Please, give the (average value ± standard error) per coating.
- 6 is not clear. Please, present the relevant data in two histograms: one concerning the average friction coefficient values, the other concerning the wear rates, per coating.
- The images in Fig. 7, as well as in Fig. 8, should be enlarged and re-arranged.
- The English language must be improved; many syntaxes and grammatical/ typing errors can be found all over the text.
Author Response
Dear Editors and Reviewers,
Thanks very much for your supervision of the reviewing process of our manuscript ID (No.: coatings-862496). We have revised the manuscript “Wear resistant of laser cladding Ti-based and WS2+hBN self-lubricating composite coating” based on comments and suggestions from the Reviews and Editors. Those comments are all valuable and very helpful for revising and improving our paper, as well as the important guiding significance to our researches. We have studied comments carefully and have made correction which we hope meet with approval. To make the reviewer comments available to all of the authors, we have repeated the comments from each of the reviewers below (in a different font). We have responded to the comments by each reviewer where appropriate and have indicated the changes made in the manuscript. We also highly appreciate the reviewers’ carefulness, conscientiousness, and the rich knowledge on the relevant research fields, since he/she has given me a number of beneficial suggestions. Below are detailed responses to comments from the editors and reviewers. Revised portion are marked in red in the paper. The page and line numbers refer to our revised manuscript submitted. The main corrections in the paper and the responds to the reviewer’s comments are as flowing.
Kind regards,
Kaiwei Liu, Hua Yan, Peilei Zhang, Jian Zhao, Qinghua Lu, Zhishui Yu
Reviewer #2: This work deals with the use laser cladding technology for elaborating composite coatings, exhibiting self-lubricating properties, onto a two-phase Ti alloy. In principle, this issue is of significant engineering importance and suitable for Coatings. However, I cannot propose the publication of the article in its present form, since it suffers at several points:
- What was the volume of the pre-deposited powder mixture ?
We are very sorry for the miss. In order to ensure the quality of the coating, we are equipped with 200g for each powder. The volume of the pre-deposited powder mixture we will add to (L. 84, page 2).
(L. 84, page 2) To ensure homogeneous mixing, the mixture was milled in a planetary ball mill for 4 h, each powder equipped with 1000g.
- Please revise the content in parenthesis (L. 100, page 3) and the rotational speed (L. 110, page 3): volume ratio HF:HCl:H2O = 1:3:46 and 100 rpm.
We agree the reviewer’s suggestion that unit professionalism, Amended in the paper as follows.
(L. 105, page 3) Lastly, the sample was etched with Keller reagent (volume ratio HF: HCL: H2O = 1: 3: 46) for 20 s.
(L. 114, page 3) All the wear test specimens were with diameter 15 mm and thickness 5 mm during 30min Dry slip friction test in which the parameter was 5.0kg normal load, the speed was 100rpm and the grinding radius was 3mm.
3. Equation (1), L. 115-116, page 3, does not appear correctly.
According to the reviewer’s suggestion, we have modified the text in 2. equation (1) and it is shown below.
(L. 120, page 3)The wear rate is expressed by the wear expressions below:
(1)
Where W, V, L and D respectively denote specific wear rate (mm3, N-1, m-1), wear volume (mm3), normal load (N) and sliding distance (m) [20].
4. Fig, 2 should be moved earlier and closer to the relevant text. Moreover, the red-coloured marks in (b)), (c) and (d) do not help the reader to follow the discussion. Especially, the marks on (c) that are combined with the data given in Table 2, are not at all clear. Moreover, the low magnification of the image does not allow the reader to understand differences of the microstructure features that could be related to differences of the elemental composition.
Thanks for the reviewer’s suggestion for revising the section of Fig.2 in our manuscript. The suggestion put forward by the reviewer are of great significance. According to the reviewer’s suggestion, we have modified the text in section 3.1 and it is shown below.
(L. 129, page 4) X-ray diffraction (XRD) analysis regions of the five samples are presented in Fig.2. The laser cladding process exhibits a non-equilibrium state of rapid solidification, thereby resulting in multiphase coexistence, and the diffraction peaks of the phase overlap the equilibrium state, so laser cladding coating is difficult to recognize all phases. All diffraction peaks are referenced based on the Joint Committee on Powder Diffraction Standards (JCPDS). As revealed from the index results, the main diffraction peak of the coating complies with the following JCPDS cards: TiN was 01-087-0632, TiW was 03-065-6898, TiS was 00-051-1329, and TiB was 00-006-0641. As revealed from the analysis of the XRD pattern, the coating largely consisted of γ-Ni, TiN, TiNi, TiW, WS2 and TiS mixture. Mixture (e.g., TiNi, TiW, and TiS) was formed by vast Ti element generated by a partially melted matrix and Ni, S, and W elements were generated by the additive deposited metal that reacted with each other in the molten pool. The diffraction peaks were clearly observed in the coating with TiN added, suggesting that the chemical durability of TiN was high. TiN had a diffraction peak at 36.7°ï¼ˆ111), 42.63°ï¼ˆ200), and 61.9°ï¼ˆ220), indicating that the composite coating was a mixed crystal grains with orientation components. Since TiN is a face-centered cubic lattice structure with a closely packed (111) direction, the energy required to grow TiN in this direction was low. Accordingly, the preferred orientation of TiN is (111)[21]. As suggested from the XRD analysis, the diffraction peak intensity of coating 2 at 39.554° was the strongest, and the diffraction peak intensity of coating 3 coating 4 and coating 5 increased with the increase in WS2 phase. It was because the intermetallic compound TiW was formed, and the diffraction peak was present at 39.554°. In the course of laser cladding, though WS2 was easily decomposed (decomposition temperature at 510 °C), the experiment used a certain method to save or regenerate into WS2. The diffraction peak of the lubricating phase hBN was clearly identified in coating 1.
Figure 2. XRD pattern of composite coating.
The cross-section microscopic morphology of composite coatings was characterized in Fig.3, containing a continuous matrix, dendrites phase, bar phase and branches crystals, and the phase structure was more refined (the grain size was smaller). Fig.3 (a) is overview cross-sectional feature of a composite coating. This figure suggests that the coating and substrate exhibited a well metallurgical combination. Fig.3(b) shows the upper region of the composite coating, the middle part of the composite coating is depicted in Fig.3(c). Contrast the upper, middle and bottom parts of the coating, upper region had a larger microstructure than the intermediate region, since the degree of supercooling and cooling rate of the upper part of the composite coating surpassed the middle region of the coating, and the critical nucleation radius of the upper part of the coating was larger, thereby achieving a larger microstructure of the upper part of the coating. The composite coating had fine dendrites distributed in the region close to the bonding zone, indicated in Fig.3 (d), and the bonding region showed planar crystal and columnar crystal growth, significantly different from the microstructure of the upper and middle regions of the coating. Since the bonding zone was close to the substrate, the temperature gradient G (G as the liquid phase temperature gradient) close to the substrate region in the initial molten pool solidification phase was significantly larger than zero; the crystallization velocity R was closed to zero, and the value of G/R was close to infinity. The planar crystal preferentially grew outward from the substrate. Subsequent temperature rise of the substrate caused G to decrease and R to increase, and then the value of G/R decreased, facilitating the growth of dendrites. Accordingly, fine dendrites were distributed near the bonding zone.
Figure 3. SEM topography of coatings: (a) macroscopic morphology, (b) upper part, (c) middle part (d) bottom part.
The chemical composition of the different phases in coating (Fig. 4) was determined with an EDS spectrometer. The distribution of the elements is listed in Table 3. The EDS study outcomes and XRD analysis suggested that the dendritic phase A in Fig. 4 primarily contained Ti (49.50%) and N (49.19%) elements, which were presumed as TiN; the gray phase B primarily covered Ti and Ni elements. It is speculated that the two phases are TiNi; the acicular phase C is mainly composed of Ti (72.09%) and S (9.91%) elements, which is supposed to be TiS; the layered substance D in the figure is primarily composed of W (13.72%); the composition of S (15.62%) is presumed to be WS2; the phase F is main containing of Ti (46.35%) and W (42.28%) elements, as presumed to be TiW. It is therefore revealed that the Ni-based self-lubricating composite coating covers both a hard phase (e.g., TiN, TiNi, TiW, and a TiS lubricating phase formed in situ). The hBN content is small and cannot be identified in EDS.
Figure 4. SEM topography of the chemical composition of the different phases in coating
5. The values of microhardness that are given with a precision of two decimals are rather confusing. Please, give the (average value ± standard error) per coating.
Considering the Reviewer’s suggestion, we have modified the sentences in section 3.2 and added some own results. The modifications are as follows:
(L. 197, Page 7) The average microhardness of the five coatings was 860 ±12.8HV0.2 (coating 1), 842±16.4 HV0.2 (coating 2), 870 ±14.5HV0.2 (coating 3), 1006 ±6.6HV0.2 (coating 4), and 926.5 ±19.8HV0.2 (coating 5), respectively.
6. 6 is not clear. Please, present the relevant data in two histograms: one concerning the average friction coefficient values, the other concerning the wear rates, per coating.
We agree the reviewer’s suggestion that the image data should be clearly displayed in the paper. The modifications are as follows:
7. The images in Fig. 7, as well as in Fig. 8, should be enlarged and re-arranged.
According to the reviewer’s suggestion, we have modified the image in Fig.7 and Fig. 8, simultaneously, we revise some mark in photograph.
Figure 7.Wear scar morphology and wear volume by 3D microscopic system of extreme field depth. (a) coating1, (b) coating 2, (c) coating 3, (d) coating 4, (e) coating 5.
Figure 8. SEM images of the surface wear morphology of the different coated samples: (a) coating 1, (b) coating 2, (c) coating 3, (d) coating 4, (e) coating 5.
8. The English language must be improved; many syntaxes and grammatical/ typing errors can be found all over the text.
We are sorry for this error. The sentence that the reviewer mentioned has been modified. we have edited the manuscript carefully. Moreover, the manuscript has been checked by Dr. Zhaoting Xiong in Loughborough University (U.K.) and he has given many suggestions for smoothing the revision. All the modifications have been marked in red.

Round 2
Reviewer 1 Report
The authors have revised the manuscript very well.
Author Response
Dear Editors and Reviewers, Thanks very much for your supervision of the reviewing process of our manuscript ID (No.: coatings-862496). We have revised the manuscript “Wear resistant of laser cladding Ti-based and WS2+hBN self-lubricating composite coating” based on comments and suggestions from the Reviews and Editors. Those comments are all valuable and very helpful for revising and improving our paper, as well as the important guiding significance to our researches. We have studied comments carefully and have made correction which we hope meet with approval. To make the reviewer comments available to all of the authors, we have repeated the comments from each of the reviewers below (in a different font). We have responded to the comments by each reviewer where appropriate and have indicated the changes made in the manuscript. We also highly appreciate the reviewers’ carefulness, conscientiousness, and the rich knowledge on the relevant research fields, since he/she has given me a number of beneficial suggestions. Below are detailed responses to comments from the editors and reviewers. Revised portion are marked in red in the paper. The page and line numbers refer to our revised manuscript submitted. The main corrections in the paper and the responds to the reviewer’s comments are as flowing. Kind regards, Kaiwei Liu, Hua Yan, Peilei Zhang, Jian Zhao, Qinghua Lu, Zhishui Yu ------------------------------------------------------------------------------------------------------- Reviewer Comments: Reviewer #1: English language and style are fine/minor spell check required. Response: We are sorry for this error. The sentence that the reviewer mentioned has been modified. we have edited the manuscript carefully. Moreover, the manuscript has been checked by Dr. Zhaoting Xiong in Loughborough University (U.K.) and he has given many suggestions for smoothing the revision. All the modifications have been marked in red.
Reviewer 2 Report
The manuscript has been extensively revised. However, some of the comments have not been taken into account:
- The first correction, concerning the volume of the powder, is rather wrong. The comment concerned the volume, and not the mass, in g (P.2, L.84).
- The marks (in red) in several images remain unclear for the reader.
Author Response

(The authors gave the same response as above.)
